# D-Dimer Assessment to Predict Pulmonary Embolism in ICU Patients with COVID-19 Pneumonia

**Jelger Louwsma** [1], **Bas Langeveld** [2], **Jacqueline M. Luyendijk** [3] and **Huub L. A. van den Oever** [1,*]

1   Intensive Care Unit, Deventer Hospital, Nico Bolkesteinlaan 75, 7416 SE Deventer, The Netherlands; jelger.louwsma@gmail.com
2   Department of Pulmonology, Deventer Hospital, Nico Bolkesteinlaan 75, 7416 SE Deventer, The Netherlands; b.langeveld@dz.nl
3   Department of Radiology, Deventer Hospital, Nico Bolkesteinlaan 75, 7416 SE Deventer, The Netherlands; j.luyendijk@dz.nl
*   Correspondence: h.vandenoever@dz.nl

**Abstract:** The value of D-dimer assessments in ICU patients with COVID-19 for the prediction of pulmonary embolism (PE) is unclear. The present study had two purposes: 1. To assess the specificity of elevated absolute D-dimer values for PE on admission to the ICU. 2. To assess the specificity of a D-dimer increment for the development of PE during an ICU stay. D-dimer values were paired with the results of a CT pulmonary angiogram (CTPA) and compared in patients with and without PE on admission. In patients without PE on initial imaging and available repeat CTPA during an ICU stay, D-dimer increments between initial and repeat imaging of patients developing PE during an ICU stay were compared with those with persistently no PE. On admission, D-dimers in patients with PE were higher than those in patients without PE (median 850 vs. 6060 µg/L; $p < 0.0001$). Using a cut-off of 9000 µg/L, the specificity for predicting PE was 100% (CI 95.3–100%). Delta D-dimer during an ICU stay was greater in patients with PE (median 7983 vs. 3815 µg/L; $p < 0.005$). Using a cut-off of 8000 µg/L, specificity was 100% (CI 79.4–100%). Strongly elevated D-dimer values on admission and marked increases in D-dimer during ICU stays have a high specificity for predicting pulmonary embolism in critically ill COVID-19 patients.

**Keywords:** COVID-19; pulmonary embolism; D-dimer





## 1. Introduction

Pulmonary embolism (PE) is a common complication of COVID-19, the respiratory disease caused by the SARS-CoV-2 virus, and is associated with an increased mortality risk in hospitalized patients [1,2]. PE has been reported in up to one-third of COVID-19 patients admitted to the ICU [1–4]. PE can be present on admission or it can occur during the course of the ICU admission. In both instances, it could help clinicians to have a marker that flags PE for consideration, especially because the signs and symptoms of PE can be masked by severe COVID-19 pneumonia. D-dimer is a biomarker used to exclude PE, but is not recommended for confirmation of the diagnosis because of the high level of false positives [5].

Elevated D-dimer levels are frequent in acutely ill individuals due to systemic inflammation and have been associated with increased mortality in COVID-19 [2,6]. As markers of disease severity, some institutions have included D-dimers in routine laboratory packages for COVID-19 patients. This way, clinicians caring for patients who are critically ill with COVID-19 may have encountered unusually high D-dimer levels without clear guidance on how to interpret them.

We conducted this study to assess the following: 1. The specificity of an elevated absolute D-dimer value for predicting PE in COVID-19 patients on admission to the ICU. 2. The specificity of a D-dimer increment in critically ill COVID-19 patients with no PE on the initial CT pulmonary angiogram (CTPA), for the development of PE during an ICU stay.

## 2. Materials and Methods

We conducted a case–control study in patients with COVID-19 pneumonia admitted to the ICU of the Deventer Hospital between August 2020 and May 2021. Because of the retrospective nature of the study, the Ethical Review board of Isala Clinics at Zwolle, Netherlands, waived the need for prior ethical approval. The research was conducted according to the principles of the World Medical Association Declaration of Helsinki.

Patients were selected if at least one CTPA was made. The decision to perform CTPA was made by the clinical treatment team.

Patients without PE on initial imaging received low-molecular-weight heparin (LMWH) in elevated prophylactic doses (nadroparine 5700 IU subcutaneously, once daily), and those with proven PE were treated with LMWH in therapeutic doses. Patients who were already on therapeutic anticoagulants before admission were excluded from the analysis. D-dimers were measured twice weekly. The medical team (including an intensivist and a pulmonologist) ordered a CTPA when the clinical status deteriorated, oxygenation decreased, end-tidal-arterial $PCO_2$ gap increased, or at the team's own discretion. Relevant patient information was collected from the electronic patient records.

D-dimer values were paired with CTPA results. The last D-dimer value before each CTPA was used for the analysis. When unavailable, the D-dimer value closest to the CTPA (maximum 2 days after CTPA) was used.

For the first research question, D-dimer values were compared between patients with PE and without PE on initial CTPA. Sensitivity, specificity, and accuracy were calculated using a standard cut-off value of 500 μg/L and an adjusted cut-off value based on our results.

For the second research question, patients with a negative first CPTA and at least one repeat CPTA were selected. Two groups were created: patients who developed PE during an ICU stay and those without PE on repeat imaging. D-dimer increment in the first group was calculated by subtracting the D-dimer value at the time of the last negative CTPA from the value at the time of the positive CPTA. In the second group, D-dimer increment was calculated by subtracting the D-dimer value at the time of the first CTPA from the value at the time of the second CPTA. In patients who underwent more than two CTPAs, the last two were used to calculate D-dimer increment (Figure 1). SPSS Statistics version 26 was used for statistical analysis. Wilcoxon sign rank test was used to compare groups.

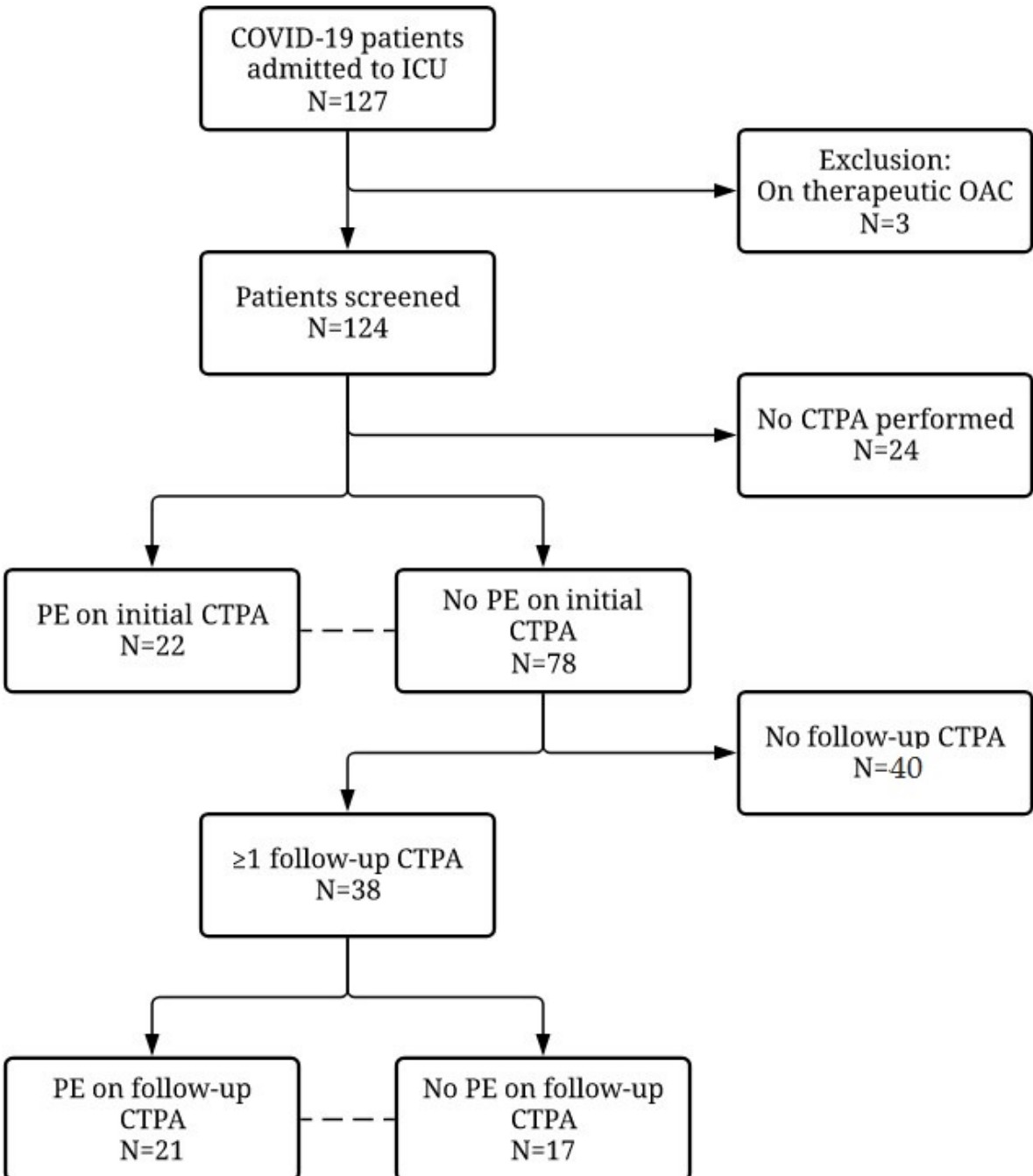

**Figure 1.** Flowchart of patient selection and inclusion. The dashed lines illustrate the groups that were compared. Abbreviations: OAC, oral anticoagulants; CTPA, computed tomography pulmonary angiogram; PE, pulmonary embolism.

### 3. Results

Between August 2020 and May 2021, when wild-type, alpha, and delta variants were predominant in the Netherlands, 124 eligible COVID-19 patients were admitted to the ICU (Table 1). CTPA was performed in 100 patients with moderate-to-high clinical suspicion of PE (decreased oxygenation, increased difference between arterial and end-tidal $pCO_2$, worsening ventilator parameters, or other clinical suspicion) before admission to the ICU; PE was diagnosed in 22 patients (17.7% of admissions). All patients suffered their first episode of COVID-19, and none of the patients were vaccinated at the time of admission. None of the patients were on extracorporeal membrane oxygenation and none of the

patients had malignant disease at the time of admission. Patients had a median of one comorbidity on the Charlson comorbidity scale.

**Table 1.** General characteristics of 124 COVID-19 patients admitted to the ICU in the first, second, and third wave of COVID-19. Results as N (%) or median (IQR). Patients with PE on initial CTPA were compared to patients without PE on initial CTPA using the Wilcoxon signed rank test. Abbreviations: CTPA, computed tomography pulmonary angiogram; PE, pulmonary embolism.

|  | All Patients (*N* = 124) | No CTPA Performed (*N* = 24) | No PE on Initial CTPA (*N* = 78) | PE on Initial CTPA (*N* = 22) | *p*-Value |
|---|---|---|---|---|---|
| Sex (male) | 84 (67.7) | 15 (62.5) | 52 (66.7) | 17 (77.3) | 0.347 |
| Age (years) | 66 (57–72) | 62 (57–72) | 66 (58–71) | 66 (56–72) | 0.913 |
| Mechanical ventilation | 84 (67.7) | 13 (54.2) | 56 (71.8) | 15 (68.2) | 0.745 |
| Mortality | 21 (16.9) | 1 (4.2) | 14 (17.9) | 6 (27.3) | 0.339 |
| D-dimers on admission (μg/L) |  |  | 850 (492–1570) | 6060 (1105–16,600) | <0.0001 |

Using a cut-off value of 500 μg/L, D-dimer values on admission were elevated in most patients, but more so in those with proven PE (Table 1). The sensitivity for PE was 95.2%, with a specificity of 27.3% (accuracy: 61.2%; area under the Receiver Operating Curve (ROC): 0.78; CI 0.66–0.90). Using a D-dimer cut-off of 9000 μg/L (see Figure 2A) increased the specificity to 100% for PE, while sensitivity decreased to 28.6% (accuracy: 61.6%).

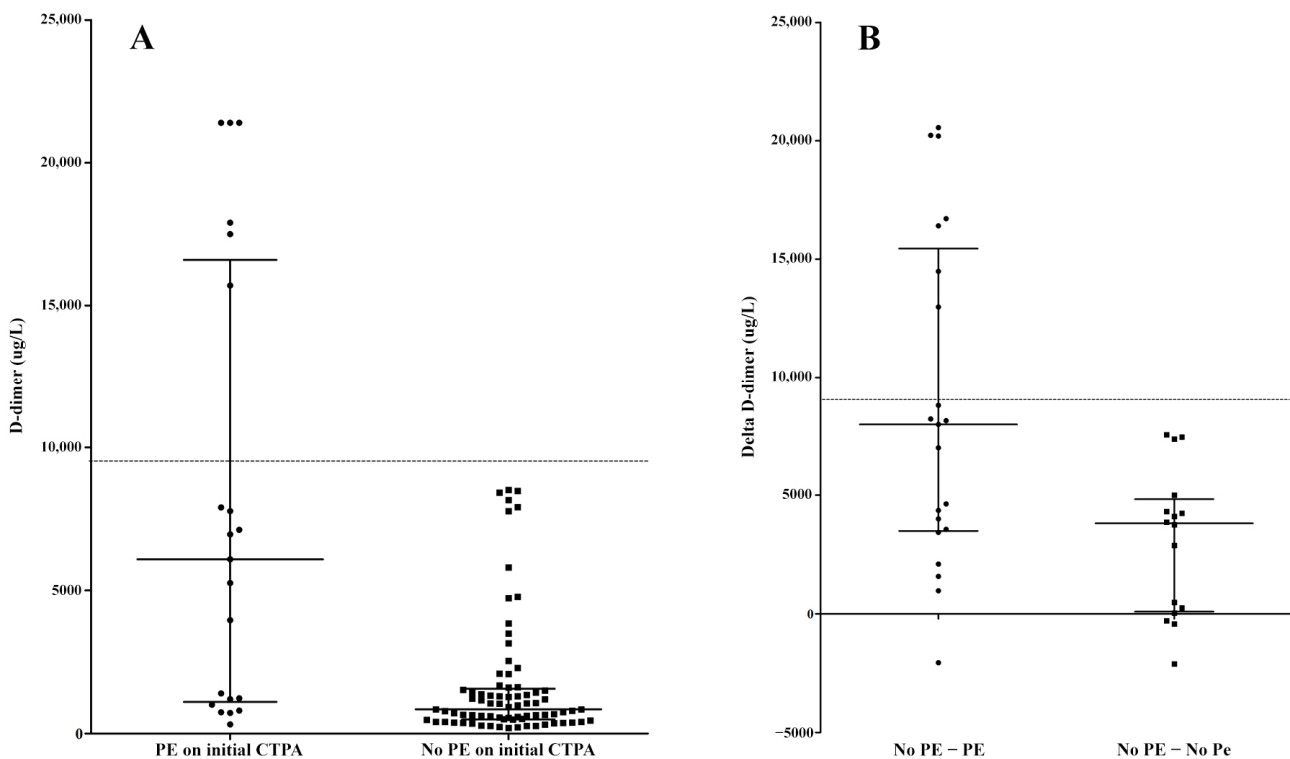

**Figure 2.** D-dimers in critically ill COVID-19 patients; data are presented as median with interquartile range. (**A**) Absolute D-dimer values of patients with and without PE on CTPA on admission to the ICU. The dashed line indicates the adjusted cut-off value of 9000 μg/L. (**B**) Delta D-dimer values of patients without pulmonary embolism on initial CTPA, who developed PE during ICU stay (indicated as "No PE–PE"), compared to patients without PE on initial CTPA and no PE on repeat CTPA during ICU stay (indicated as "No PE–No PE"). The dashed line indicates the cut-off values of 8000 μg/L for delta D-dimer. Abbreviations: CTPA, computed tomography pulmonary angiogram; PE, pulmonary embolism.

In 38 patients without PE on initial CTPA, repeat imaging was obtained during the ICU stay because of renewed clinical suspicion. PE was diagnosed in 21 patients, leading to an overall incidence of PE of 34.7% of ICU admissions. Admission D-dimer values in patients who developed PE during the ICU stay were comparable to those in patients who did not (median: 789, IQR: 503–5295 µg/L vs. 993, IQR: 378–3198 µg/L; $p = 0.87$). At repeat imaging, delta D-dimer was positive in both groups, but more so in those with PE (median: 7983, IQR: 3497–15,450 µg/L vs. 3815, IQR: 94–4830; $p < 0.005$; Figure 2B). An increase in D-dimer of 8000 µg/L or higher had a specificity of 100% for the development of PE, with a sensitivity of 47.6% (accuracy: 73.8%; area under the ROC: 0.76; CI 0.60–0.91).

## 4. Discussion

The results of this study showed that the D-dimer levels were elevated in most critically ill patients with COVID-19, but more in those with proven PE. In our study population of critically ill patients with COVID-19 with a moderate-to-high clinical suspicion of PE, the D-dimer levels on admission of 9000 µg/L had a specificity of 100% for predicting PE. In the same population, a D-dimer increment of 8000 µg/L or higher at any time during the course of the ICU admission predicted PE with a specificity of 100%.

Of all the patients admitted to the ICU with COVID-19 pneumonia, more than one-third (34.7%) had a diagnosis of PE at some time during their ICU stay. About half of these patients had PE on admission, and the other patients developed PE during an ICU stay. The high incidence of PE in severely ill COVID-19 patients corresponds to other studies (Table 2). Therefore, PE has been established as a frequent complication in critically ill COVID-19 patients.

When COVID-19 patients are admitted to the ICU, D-dimer levels above the normal cut-off value of 500 µg/L are a common finding [7,8]. This finding has been recognized as a risk factor for mortality [9], and it has been suggested that D-dimer levels elevated far above their normal value have a strong predictive value for venous thromboembolism in ICU and non-ICU patients [10]. In COVID-19 patients, the specificity and positive predictive value of D-dimer for PE benefit from raising the diagnostic threshold of the D-dimer test [11]. Most authors who investigated the diagnostic value of D-dimer testing have accomplished this.

Depending on the purpose of the study, the new cut-off value can be chosen to be relatively low to improve the sensitivity of the test and hence the predictive value of a negative test result, or the cut-off can be chosen to be relatively high to improve the specificity and the positive predictive value. In Table 1, the sensitivity and specificity of a low (500 µg/L) threshold and a high threshold (9000 µg/L) for our data are presented.

A literature search revealed three studies that were performed in COVID-19 ICU populations and reported the sensitivity and specificity of D-dimers alone (not as part of a multi-item scale) for the prediction of PE alone (not for venous thromboembolism as a whole) [2,12,13]. Table 2 summarizes the main findings of these studies combined with the present study. In all the studies, patients received thromboprophylaxis throughout their stay in the ICU, though the regimens varied. The incidence of PE in our study matched with previous findings, though in a recent meta-analysis, the frequency of PE in patients with COVID-19 admitted to the ICU based on 22 publications was estimated at 48.6% [10]. All studies summarized in Table 2 used adjusted cut-off values for PE diagnosis.

When laboratory results are reviewed during daily ward rounds, D-dimer increments may attract attention. We found no other studies that investigated the diagnostic value of D-dimer increments in ICU patients during the course of admission, though the data presented by Van den Berg et al. illustrated that D-dimer values tended to increase remarkably in the days prior to a positive CTPA [2]. Our results demonstrated that in severely ill COVID-19 patients, D-dimer increments higher than 8000 µg/L indicated that PE was very likely. Therefore, these results might help clinicians with the decision to order definitive radiology when faced with strongly elevated D-dimers in a severely ill COVID-19 patient. Moderately increased D-dimer values should be interpreted with caution, but a marked increment could aid the clinician in the early recognition of PE in COVID-19 patients in the ICU.

**Table 2.** Summary of studies reporting sensitivity and specificity of D-dimer for PE in patients with COVID-19 admitted to the ICU. Abbreviations: AUC, area under the curve; bd, twice daily; ICU, intensive care unit; IU, international units; od, once daily; PE, pulmonary embolism.

| Study | Number of ICU Patients | Study Design | Type of Thrombo-Prophylaxis | Incidence of PE in ICU | D-Dimers of Patients without PE (µg/L) | D-Dimers of Patients with PE (µg/L) | Cut-Off for PE Diagnosis (µg/L) | Sensitivity of D-Dimer (%) | Specificity of D-Dimer (%) | AUC of D-Dimer |
|---|---|---|---|---|---|---|---|---|---|---|
| Faqihi [12] | 160 | Cohort | Enoxiparin: <50 kg, 20 mg od; 50–100 kg, 40 mg od; 100–150 kg, 40 mg bd; >150 kg, 60 mg bd | 34.4 | 2800 (1200–5500) | 5100 (2200–9100) | 3000 | 74.5 | 95.1 | ns |
| Taccone [13] | 40 | Cohort | Enoxiparin: 4000 IU od; later 4000 IU bd | 32.5 | 2302 (1327–5750) | 8280 (5976–11,483) | 3647 | 75 | 92 | 0.90 |
| van den Berg [2] | 76 | Cohort | Dalteparin: <100 kg, 5000 IU od; >100 kg, 5000 IU bd | 34.2 | 2785 (973–3298) | 3190 (1600–11,545) | 8460 | 74 | 86 | 0.82 |
| Present study | 100 | Case–control | Nadroparin: 5700 IU od | 34.7 | 850 (492–1570) | 6060 (1105–16,600) | 9000 | 28.6 | 100 | 0.78 |

Due to the case–control study design, the true incidence of elevated D-dimers in our population was not known. Therefore, positive and negative predictive values could not be determined. However, with 100% specificity and a disease incidence of 34.7%, the positive predictive value is likely to be high. There is concern that test accuracies derived from case–control studies might be falsely high due to differences in selection criteria and asynchrony in time course between groups [14]. However, in our study, case and control patients came from the same population and had a similar clinical suspicion of PE and similar monitoring periods, which reduced the likelihood of either form of bias. Moreover, the AUC for our data (0.78) did not suggest a falsely high accuracy when compared to the other studies in similar populations of COVID-19 patients (Table 2). Areas under the ROC between 0.75 and 0.80 are generally not considered good enough for diagnostic purposes. Therefore, the conclusion of our results is not that D-dimer is a suitable test to diagnose PE or that it can replace radiological imaging, but rather that routine D-dimer assessments may help to identify patients with a high a priori chance of having PE on CTPA.

In outpatient or emergency ward situations, a formal assessment of pre-test likelihood (e.g., using Wells' or Geneva scores) for PE can improve the diagnostic accuracy of a D-dimer test [15]. In our population, these tests were not performed because they lacked validation in a critically ill cohort. Various elements of these scores (such as tachycardia, immobilisation, and "no alternative diagnosis") might be difficult to interpret in critically ill, ventilated patients. In one study, in a cohort of patients hospitalised with COVID-19, patients with PE had the same proportions of positive Wells' scores as patients without PE (both 25%), which indicated that Wells' scores are not a useful screening tool in severe COVID-19 [16]. Rindi et al. reviewed the literature on several predictive scores in COVID-19 patients and concluded that none of the "old" scoring systems had predictive ability [17]. The newly developed CHOD score, specifically designed to predict PE in hospitalised COVID-19 patients, showed some promise, but it leaned heavily on D-dimer as one of its major components [18]. Pre-test likelihood scores were not used in our study for these reasons, but also because COVID-19 requiring ICU admission itself already came with a pre-test likelihood of 34.7% (our data).

Age is known to change the accuracy of the D-dimer test for PE diagnosis due to a higher level of D-dimer as individuals age, and a higher threshold of D-dimer has been shown to increase the specificity of D-dimer to diagnose PE in older patients [19]. The median age of the PE patients was the same as for patients without PE, so it is unlikely that age would have confounded our general conclusions. However, it is possible that future studies will demonstrate that for the prediction of PE in severe COVID-19, different cut-off values are applicable to different age groups. Larger studies will be needed to show a possible effect of age on the accuracy of D-dimers in this population. Our study had insufficient power to identify subgroup differences.

Data of this study were collected during the first year of the COVID-19 pandemic, when wild-type, alpha, and delta variants were prevalent worldwide. This may limit the generalisability to other SARS-CoV-2 variants. In particular, if later variants come with a lower incidence of PE, this might decrease the predictive value of D-dimer assessment.

Another possible weakness of this study was that other forms of thrombosis as a possible source of D-dimers were not addressed. Concepts about the pathogenesis of PE in COVID-19 have shifted away from venous thromboembolism as the primary mechanism. It has been recognized that COVID-19 patients are prone to develop pulmonary microthrombosis, a disease process in which local inflammation involves the pulmonary endothelium and enhances thrombus formation at the site of severe disease [20]. However, venous thrombosis could be an alternative explanation for the overall elevation of D-dimers in this population, as well as the handful of outliers in Figure 2A, with high D-dimers, yet without PE. It is therefore unlikely that a D-dimer test could replace radiographic imaging in the diagnosis of PE.

## 5. Conclusions

Routine measurement of D-dimer might aid clinicians in diagnosing PE in patients with COVID-19 pneumonia admitted to the ICU. Both an absolute D-dimer value > 9000 μg/L, and an increase in D-dimers > 8000 μg/L during the course of an ICU admission were strongly associated with PE.

*Take home message:* Strongly elevated D-dimers on admission or a marked increases in D-dimers during an ICU stay have strong predictive values for pulmonary embolism in critically ill COVID-19 patients and may aid clinicians in choosing the appropriate diagnostic follow-up.

**Author Contributions:** J.L.: data extraction, analysis and writing of the paper. B.L.: conceptualization of the study, data analysis. J.M.L.: data analysis. H.L.A.v.d.O.: conceptualization of the study, analysis and writing of the paper. All authors have read and agreed to the published version of the manuscript.

**Funding:** This research received no external funding.

**Institutional Review Board Statement:** Because of the retrospective nature of the study, the Ethical Review board of Isala Clinics at Zwolle, Netherlands, waived the need for prior ethical approval. The research was conducted according to the principles of the World Medical Association Declaration of Helsinki.

**Informed Consent Statement:** Patient consent was waived due to the retrospective nature of the study.

**Data Availability Statement:** The dataset generated and analysed during this study is available from the corresponding author on request.

**Conflicts of Interest:** The authors declare no conflict of interest.

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
