# Peer review of "D-Dimer Assessment to Predict Pulmonary Embolism in ICU Patients with COVID-19 Pneumonia"

_covid, doi:10.3390/covid3090095_

Round 1
Reviewer 1 Report
Dear Authors!
I read your paper with an interest as it touches a very important point of clinical significance of D-Dimer level in COVID-19 pneumonia patients. The data you presented are interesting and may help physicians in their practice.
I have only two minor remarks.
1. In a flowchart presented on figure 1 I did not fully understand how the group of 78 patients with no PE on initial CTPA was divided. If you sum 44 and 38 in subsequent boxes it results in 82, but not in 78.
2. As a limitation of the study you have to mention that data were obtained from the very first cohorts of COVID19 patients. So, data cannot be generalized on patients with COVID-19 variants that spread later.
None
Author Response
Dear Reviewer,
Thank you for your remarks. It is a reassurance to know that you have read our manuscript carefully.
Regarding the first point, you are absolutely right. We have adjusted Figure 1 so that of 78 patients with no PE on initial CTPA, 38 patients had follow-up CT scanning, and 40 did not.
Your second point is also clear. We already reported in our results in what period our data were collected and which serotypes were prevalent in these patients. We have adjusted the Discussion to include this as a limitation to the generalisability of our conclusions.
Best Regards,
Huub van den Oever
Reviewer 2 Report
The authors present a retrospective case-control study on D-dimer assessment in critically ill COVID-19 patients and its predictive value for pulmonary embolism. This manuscript is well written and has no major conceptual flaws. The authors are aware of the limitations of the study and communicate them transparently.
Other manuscripts have answered similar questions, thus, the paper itself does not carry a lot of novelty. Nevertheless, the article has clinical relevance, and as populations in such conditions are small, it is important to confirm such results.
D-dimer is known to be high in COVID-19 patients, which does not necessarily correlate with venous thrombosis. Therefore, its clinical value has been questioned. The present article has in its central message an important value: very high D-dimer values or very high increases during in-patient care should raise suspicion for PE. And in this cases, D-dimer should not be ignored. Nevertheless, the cut-off value used by authors is purely speculative and cannot be translated into clinical practice. Therefore, these must be interpreted with caution.
Even though the authors describe that tools for pre-test probability can be inefficacious for ICU-Patients, I believe it would be of clinical interest to point out if patients presenting with PE had any kind of clinical deterioration prior to CTPA. As mentioned by the authors, the sensitivity of the test is drastically reduced by establishing really high cut-off levels, thus clinical suspicion must be considered. Could the authors describe if there were any parameters that influenced the decision for performing a CTPA?
Other factors may influence D-Dimer, such as secondary infections or malignancies. It would be important to describe if these were different between groups with or without pulmonary embolism, as they could mask a “true increase” due to VTE.
Were any patients under ECMO treatment?
I would expect D-dimer to fall through time in patients who are recovering from infection. In these cases, do you think that the variation of D-dimer levels between CTPAs is adequate? Maybe at admission, a more active disease could have been responsible for the high D-dimer, but later on, the elevations could be due to thrombosis. If the values were lower in-between, it is possible that your variation between CTPAs will not be the most sensitive evaluation to detect an event. I believe it would be important to show the dynamics of D-Dimer for individual patients in both groups (e.g. with a Spaghetti-Plot) to understand if the dynamics could provide more information about the risk of developing an event.
Author Response
Dear Reviewer,
Thank you for reviewing our manuscript. It is reassuring to know that our paper has been read with interest.
Your first question was about parameters that influenced the decision to make a CTPA. We must emphasize that our data were collected from a real-world situation, without protocols to aid in the decision-making. In the Methods we described that CTPA was made at the clinicians discretion. However, typical indications for CT scanning of COVID-19 patients in our ICU were decreased oxygenation, increased difference between arterial and end-tidal pCO2, and worsening ventilator parameters. We have added this to the general patient descriptions in the Results section.
As you rightly pointed out, other patient factors (superinfections or malignacies) or invasive treatments (ECMO) could influence the incidence of PE. We have added remarks about concomitant diseases and ECMO to the Results section.
Your last remark concerned the natural course of D-dimer in the early phase of the disease, and you suggested that D-dimers would tend to fall during the course of the disease. However, we think that during the period that the second or third CTPA's were made, most patients were still in the acute phase of the infection, with high ventilatory requirements, high inspiratory oxygen fractions, and indeed a deterioration that prompted another CT scan. We are not sure if a decrease in D-dimers was to be expected in that phase. In fact, our data suggest the opposite: Even in the group with no PE on the follow-up CT scan, delta D-dimer was positive in all but three cases. Figure 2B suggests that it was not the direction of the delta d-dimer, but rather the magnitude of the positive delta D-dimer that indicated an increased risk of PE. In most patients the acute phase of the infection was a rather turbulent period, that showed much variety in duration and also in the nature of complications. Almost half of the patients with a CTPA on admission had a second CTPA at some time, but some had three or four CTPA's during this period. We think this would reduce the usefulness of a spaghetti plot. Therefore, we did not add a spaghetti plot to the Results.
Again, thank you for commenting on our manuscript.
Best Regards,
Huub van den Oever